# Averting an Unnecessary Revision of a Roux-en-Y Hepaticojejunostomy by Surgically Creating an Access Point for the Endoscopic Assessment of the Anastomosis: A Report of a Case

**DOI:** 10.3390/medicines10050031

**Published:** 2023-05-11

**Authors:** Dimitrios Symeonidis, Ismini Paraskeua, Athina A. Samara, Effrosyni Bompou, Alexandros Valaroutsos, Maria P. Ntalouka, Dimitrios Zacharoulis

**Affiliations:** 1Department of Surgery, University Hospital of Larissa, 41100 Larisa, Greece; 2Department of Anesthesiology, University Hospital of Larissa, 41100 Larisa, Greece

**Keywords:** primary sclerosing cholangitis, Roux-en-Y hepaticojejunostomy

## Abstract

**Introduction:** Primary sclerosing cholangitis sets the scene for several pathologies of both the intrahepatic and the extrahepatic biliary tree. Surgical treatment, when needed, is almost unanimously summarized in the creation of a Roux-en-Y hepaticojejunostomy, a procedure with a relatively high associated failure rate. **Presentation of case**: A 70-year-old male, diagnosed with primary sclerosing cholangitis, was submitted to a Roux-en-Y hepaticojejunostomy due to a dominant stricture of the extrahepatic biliary tree. Recurrent episodes of acute cholangitis dictated a workup in the direction of a possible stenosis at the level of the anastomosis. The imaging studies were inconclusive while both the endoscopic and the transhepatic approach failed to assess the status of the anastomosis. A laparotomy, with the intent to revise a high suspicion for stenosis hepaticojejunostomy, was decided. Intraoperatively, a decision to assess the hepaticojejunostomy prior to the scheduled surgical revision, via endoscopy, was made. In this direction, an enterotomy was made on the short jejunal blind loop in order to gain luminal access and an endoscope was propelled through the enterotomy towards the biliary enteric anastomosis. **Results:** The inspection of the anastomosis under direct endoscopic vision showed no evidences of stenosis and averted an unnecessary, under these circumstances, revision of the anastomosis. **Conclusions:** The surgical revision of a Roux-en-Y hepaticojejunostomy is a highly demanding operation with an increased associated morbidity, and it should be reserved as the final resort in the treatment algorithm. An approach of utilizing surgery to facilitate the endoscopic assessment prior to proceeding to the surgical revision of the anastomosis appears justified.

## 1. Introduction

Benign or malignant biliary strictures can both represent indications for the resection of a part of the common bile duct. While the indication for resection is relatively straightforward when a malignant obstruction of the extrahepatic biliary tree is the case, there are benign strictures, not amenable to endoscopic treatment, requiring surgical intervention, as well. Inflammatory conditions, such as chronic pancreatitis, biliary calculi, or recurrent biliary tract infections, can have a stricturing effect on the common bile duct [1]. In addition, primary sclerosing cholangitis (PSC) can be the cause of multiple strictures throughout the biliary tree; however, the vast majority of benign biliary strictures are iatrogenic, usually associated with either an open or laparoscopic cholecystectomy [2,3]. If not properly treated in a timely fashion, several complications may occur [4].

Roux-en-Y hepaticojejunostomy (RYHJ) is currently the most commonly utilized procedure for the establishment of biliary–enteric continuity after any type of resection of the extrahepatic biliary tree. Technically, a long jejunal hepatic limb, of up to 75 cm and no less than 40 cm, should be the goal in order to decrease the risk for enterobiliary-reflux-causing cholangitis. The literature reports a rise the efficiency of the procedure of up to 90% [2,5]; however, the most dismal long-term complication associated with the procedure is the development of stenosis with an incidence ranging from 5 to 38% [6]. As long as the diagnosis of stenosis is confirmed, treatment should be prompt and definite in order to prevent the dismal consequences of the obstructed bile outflow, such as choledocholithiasis, recurrent cholangitis, liver abscess formation, secondary biliary cirrhosis, and ultimately, portal hypertension [7]. First line treatment options include the dilation or the stenting of the stricture via either the transhepatic route or endoscopy [8,9]. Along with interventional radiology and endoscopy, surgery is also an integral part of the treatment algorithm as it is required, usually as the last resort, in about 25% of patients [8,9].

Although the endoscopic approach is preferred for accessing the biliary tree, as it is less traumatic and associated with fewer complications compared to the transhepatic route, it is not always feasible, especially in patients with surgically altered anatomy. Recently, endoscopic techniques have been developed greatly, and experienced endoscopists equipped with the proper apparatus can access, diagnose and perform therapeutic interventions in the majority of failed RYHJ cases [10]; however, limitations still apply and unreachability through endoscopy RYHJ is not an extremely rare clinical scenario. Modifications of the surgical technique have been proposed in order to render the anastomosis more easily accessible through endoscopy, such as performing the RYHJ using a short-limb Roux-en-Y reconstruction or the establishment of a subcutaneous access loop [11,12].

In the present report, we present the interesting case where the development of a surgical access loop, not created during the initial operation, averted an unnecessary revision of an otherwise flawless hepaticojejunostomy.

## 2. Presentation of the Case

A 70-year-old male with a known history of PSC presented to the emergency department with symptoms consistent with acute cholangitis. Right upper quadrant abdominal pain combined with a fever of up to 39 degrees Celsius and increased bilirubin levels of 4.5 mg/dL established the diagnosis. The transabdominal ultrasound confirmed the presence of dilated intrahepatic bile ducts. Three weeks ago, an additional episode of acute cholangitis was treated conservatively with antibiotics alone. From his past medical history, an RYHJ was performed five years ago, as endoscopic (via ERCP) attempts were unsuccessful in catheterizing and stenting the symptomatic dominant stricture of the extrahepatic biliary tree.

Based on the patient’s relevant surgical history, an investigation towards a possible stenosis at the level of the enteric biliary anastomosis was initiated. An MRCP, which was performed three days after admission, showed a marginal dilation of the intrahepatic bile ducts while the common bile duct was found dilated 10 mm in diameter. The study however was inconclusive regarding the actual status of the anastomosis. A direct visual assessment of the anastomosis under endoscopy was then decided; however, most probably due to a long-limb Roux-en-Y reconstruction, the anastomosis was not reachable through the endoscopy. Similarly, the percutaneous transhepatic approach, which was subsequently attempted, proved unsuccessful. The marginally dilated intrahepatic bile ducts as well as the stiff texture of the liver parenchyma on the background of PSC might have contributed to the percutaneous transhepatic cholangiography failure.

The patient, though on intravenous antibiotics, remained febrile with constantly elevated biochemical markers of cholestasis throughout his admission. Based on the high level of suspicion for a possible stenosis at the level of the hepaticojejunostomy, a surgical intervention was decided. A midline laparotomy, on the patient’s previous incision, with the intent to surgically revise the hepaticojejunostomy was undertaken. During laparotomy, dense adhesions associated with the previous operation were encountered. Extensive adhesiolysis, however, enabled the identification of the anatomy, mainly in regard to the configuration of the Roux-en-Y limbs (Figure 1). However, the detailed macroscopic assessment and the manual palpation of the anastomosis did not reveal any obvious evidences of stenosis (Figure 1).

A decision to assess the hepaticojejunostomy via endoscopy prior to proceeding to the scheduled revision was made intraoperatively. In this direction, an enterotomy was performed on the short jejunal blind loop in order to gain luminal access, and an endoscope was propelled through the enterotomy towards the biliary enteric anastomosis (Figure 2). The inspection of the anastomosis under direct vision showed no evidence of stenosis and averted an unnecessary, under these circumstances, revision of the anastomosis (Figure 3 and Figure 4). The endoscope was withdrawn and the enterotomy was finally closed in a two-layer fashion with absorbable 3-0 polyglycolic acid sutures.

The postoperative period was relatively uneventful. The patient’s septic profile improved gradually during the first postoperative week. Only biochemical markers of cholestasis, i.e., alkaline phosphatase and gamma-glutamyl transferase serum levels, remained mildly elevated on discharge.

## 3. Discussion

PSC is a rare disorder characterized by multi-focal bile duct strictures and progressive liver disease. Up to 80% of patients have concomitant inflammatory bowel disease, and most of the patients will ultimately require a liver transplantation [13]. Approximately 50% of patients have symptoms such as fatigue, pain and pruritus [14]. At diagnosis, bilirubin levels are usually within the normal range but tend to increase as the disease progresses [15]. Ultrasound imaging may reveal bile duct wall thickening and dilation. Up to 40% of patients have gallbladder abnormalities such as wall thickening and cholelithiasis [16]. Cholangiography can identify strictures and sacular dilations of the proximal bile duct [15]. In addition, up to 10% of patients will be diagnosed with a cholangiocarcinoma [15]. Apart from the end-stage liver disease, poor quality of life, caused by severe refractory symptoms from recurrent cholangitis and torturous pruritus, and unresponsiveness to standard therapy may qualify patients for liver transplantation [17]. Indications for surgical intervention represent the development of cholangiocarcinoma and the development of dominant strictures of the extrahepatic biliary tree not amenable to endoscopic treatment.

In regard to the surgical treatment of biliary strictures, a resection followed by an RYHJ is the procedure of choice. Several benign or malignant conditions, such as an iatrogenic bile duct injury, PSC presenting with a dominant extrahepatic stricture or even a cholangiocarcinoma, are among the indications for this certain type of reconstruction following a common bile duct resection. Though adequately effective, this procedure has a non-negligible failure rate [9,11]. Stenosis has major long-term complications associated with significant morbidity. The diagnosis of this dismal complication is usually straightforward as the presence of certain clinical signs and laboratory test results will dictate the appropriate diagnostic imaging work up; however, problems might arise in cases where PSC is the cause of the biliary strictures. In this setting, defining whether patients’ symptoms could be attributed to a stricture at the level of the anastomosis or to the intrahepatic obstruction on the flow of bile due to concomitant intrahepatic strictures or lithiasis could be a challenging task. Similar to our case, conventional magnetic resonance imaging often fails to discriminate these two different entities. Ideally, the use of functional imaging, such as hepatobiliary scintigraphy, could prove helpful when conventional anatomical imaging cannot effectively answer the diagnostic questions [18]. A hepatobiliary iminodiacetic acid scan (HIDA scan) can assess the pattern of biliary flow in these challenging cases with persistently dilated biliary ducts as a result of chronic inflammation and obstruction [18]. In our case, the results of a HIDA scan could, at least theoretically, alter the treatment strategy; however, this certain service is not available in our department. Transferring the patient to a location that can perform hepatobiliary scintigraphy studies was an option that was precluded by the patient’s healthcare insurance logistics.

Recently, the indications of ERCP as a treatment option for biliary strictures have been expanded with the use of balloon enteroscopes to include patients with Roux-en-Y reconstruction [19]; however, ERCP in such patients is technically challenging because of the frequent inability of the endoscope to reach the blind end, i.e., either the papilla of Vater or the bilioenteric anastomosis, depending on each individual case scenario. Balloon enteroscopes can be divided into long-type enteroscopes, developed to examine and treat small bowel conditions, and short-type enteroscopes, purposed to perform, apart from endoscopy, ERCP in patients with surgically altered anatomy [20]. Most conventional ERCP devices can be used through a short-type enteroscope but not through the long-type enteroscope due to the length of the scope; however, even with the use of a short-type enteroscope, the success rates for reaching the blind end are satisfactory [20]. Problems might arise in patients with a Roux-en-Y reconstruction without gastrectomy and in the presence of disseminated peritoneal malignancies [20]. In this setting, a balloon endoscopy represents the first-line diagnostic and treatment option. Therefore, the applicability of the technique presented in the present report is limited to settings without access to these advanced endoscopic techniques and in cases where endoscopy, although available, fails to alleviate the obstruction.

In the present report, we present the interesting and inventive way of averting an unnecessary revision of an RYHJ. More specifically, as both contemporary endoscopy and the percutnaneous transhepatic approach failed in accessing a highly suspicious region for stenosis RYHJ, a decision to surgically revise the anastomosis in a patient with a past medical history of PSC was made; however, intraoperatively, a surgical access point for a proper endoscopic assessment was created. A conventional endoscope was propelled through an enterotomy on the blind jejunal loop, enabling the complete inspection of the anastomosis under direct endoscopic vision.

In general, the use of surgery in order to facilitate the access to an inaccessible to conventional endoscopy hepaticojejunostomy has been previously described in the literature. In 1995, Asbun et al. published the outcomes of three patients with recurrent cholangitis submitted to operative enteroscopy due to the inability of conventional enteroscopy to access the RYHJ [21]. In two of these cases, the anastomosis was revised endoscopically by excising the redundant scar tissue. Similar to our case, in the third patient, the anastomosis was found patent and no further interventions were undertaken [21]. During the last two decades, several case reports have been published reporting the feasibility of transjejunal ERCP in cases of distorted upper gastrointestinal tract anatomy due to previous operation [22,23,24,25,26,27,28,29,30] (Table 1). In the vast majority of these cases, a laparoscopic-assisted approach proved adequate in performing the desired access point for the endoscope onto the jejunal wall [22,23,24,25,26,27,28,29]. The increasing expertise of the surgical community in advanced laparoscopic procedures has unavoidably rendered the laparoscopic approach as the gold standard approach for these challenging clinical scenarios, especially in cases where endoscopy is highly likely to be both the diagnostic and the definite therapeutic modality.

To our knowledge, the reports by Arzaga et al. [22] and Mansor et al. [27] are the only ones available that assessed the feasibility and efficacy of ERCP via intraoperative enteroscopy on the background of a previous RYHJ, i.e., a clinical scenario of increased likelihood for an endoscopic treatment failure. Although in both of these reports the definite treatment was provided by the use of endoscopic maneuvers alone, in such cases a surgical revision of a hepaticojejunostomy might be required. A traditional open approach is considered the approach of choice when a revision of a hepaticojejunostomy is the case. In our case, performing the procedure laparoscopically could have also been an option; however, we chose the traditional open approach because the initial indication for surgery was a revision of a failing hepaticojejunostomy, and we did not plan to perform an intraoperative enteroscopy. The decision for an enteroscopy was made intraoperatively, only after a macroscopic assessment of the anastomosis.

In conclusion, PSC sets the scene for several pathologies of both the intrahepatic and the extrahepatic biliary tree. Surgical treatment, when needed, is almost unanimously summarized in the creation of an RYHJ, a procedure with a relatively high associated failure rate. Endoscopic or transhepatic routes are the preferred first-line approaches for diagnosing and treating possible long-term complications of the procedure, mainly stenosis. The surgical revision of the anastomosis is a highly demanding operation of increased associated morbidity, and it should be reserved as the final resort in the treatment algorithm in these patients. An approach of using surgery to facilitate the endoscopic assessment prior to proceeding to the surgical revision of the anastomosis appears justified.

## Figures and Tables

**Figure 1 medicines-10-00031-f001:**
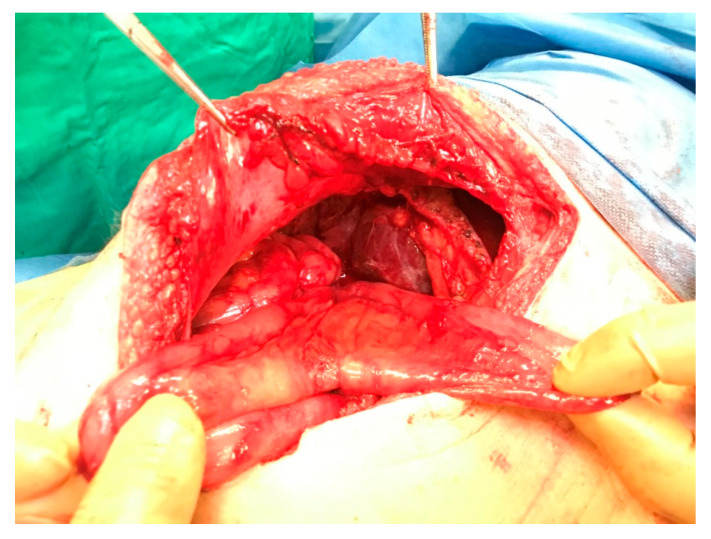
Image showing the two loops (blind and afferent loop) of the hepatic limb of the RYHJ.

**Figure 2 medicines-10-00031-f002:**
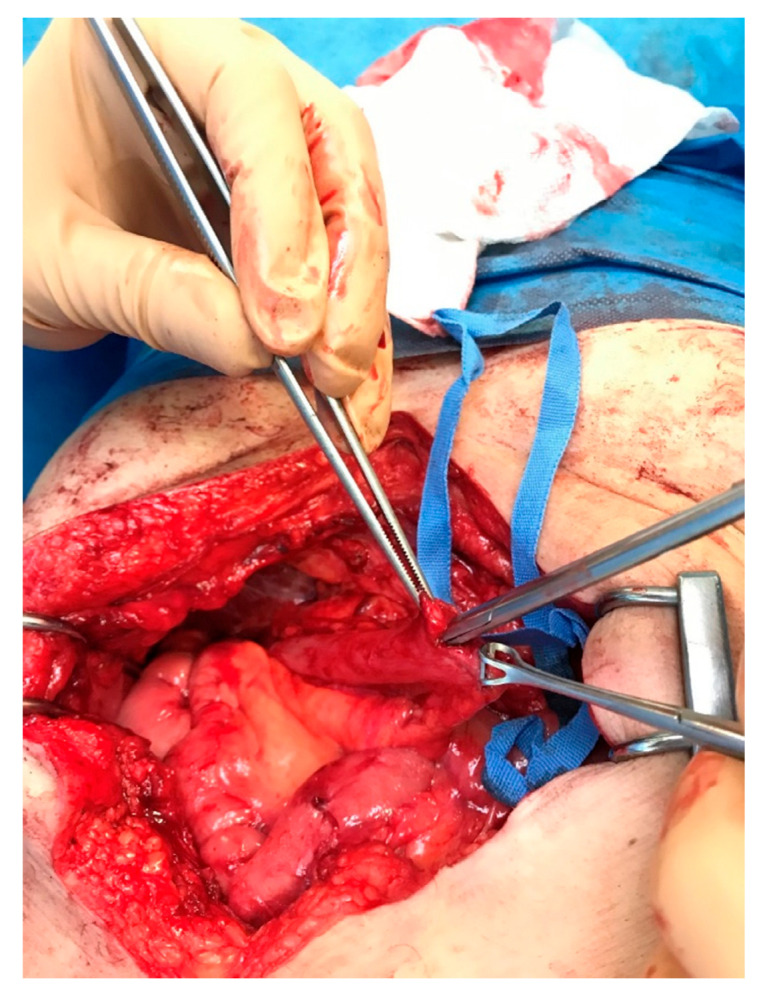
The endoscope is propelled through the enterotomy and towards the anastomosis.

**Figure 3 medicines-10-00031-f003:**
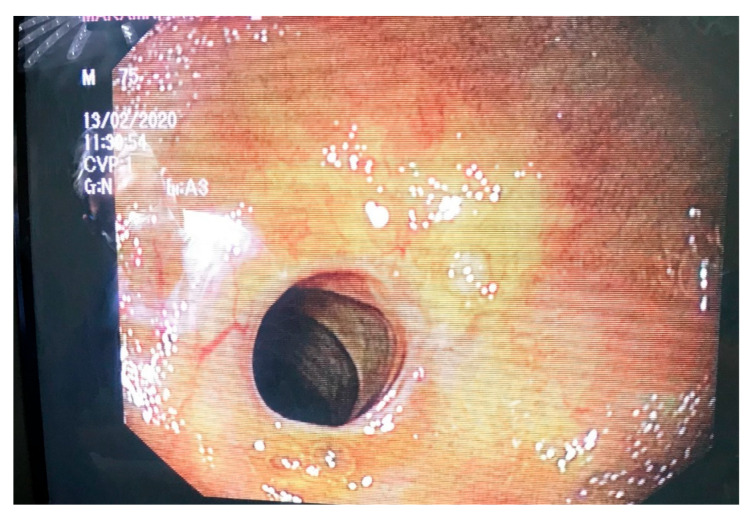
The endoscopic appearance of the anastomosis.

**Figure 4 medicines-10-00031-f004:**
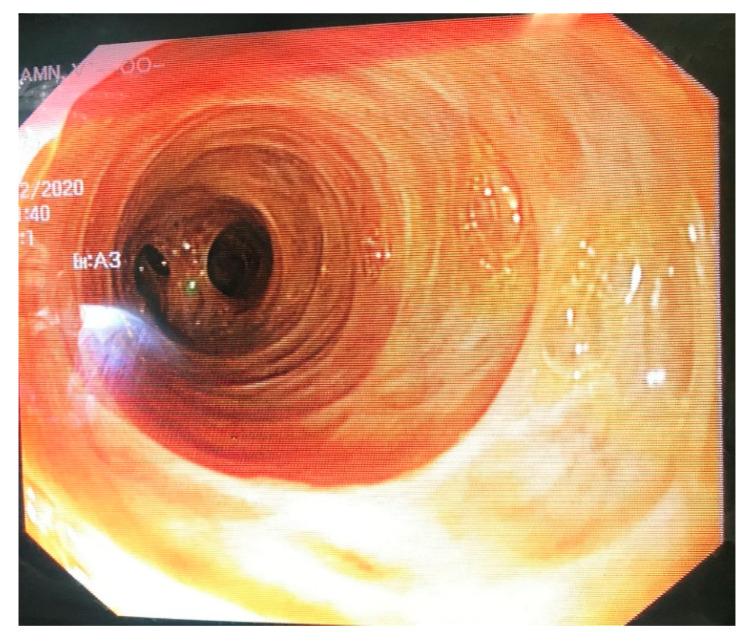
Endoscopic image of the lumen of the common hepatic duct.

**Table 1 medicines-10-00031-t001:** Literature reports of operative enteroscopy.

Author	Year	No of Patients	Previous Surgery	Diagnosis	Surgical Approach
Arzaga [22]	2019	1	Roux-en-Y hepaticojejunostomy	Intrahepatic lithiasis	Laparoscopic-assisted
Dalmonte [23]	2019	1	Roux-en-Y gastric bypass	Common bile duct stones	Laparoscopic-assisted
Mita [24]	2018	1	Subtotal gastrectomy with Roux-en-Y gastrojejunostomy	Common bile duct stones	Laparoscopic-assisted
Marchesini [25]	2017	2	Duodenal switch	Common bile duct stones	Laparoscopic-assisted
Surdeanu [26]	2016	1	Roux-en-Y gastric bypass	Common bile duct stones	Laparoscopic-assisted
Mansor [27]	2015	1	Roux-en-Y hepaticojejunostomy	Intrahepatic lithiasis	Laparoscopic-assisted
Lopes [28]	2010	1	Partial gastrectomy with Roux-en-Y gastrojejunostomy	Sphincter of Oddi dysfunction	Laparoscopic-assisted
Saleem [29]	2010	1	Subtotal gastrectomy with Roux-en-Y gastrojejunostomy	Pancreaticopleural fistula	Laparoscopic-assisted
Mergener [30]	2003	1	Subtotal gastrectomy with RY gastrojejunostomy	Biliary pancreatitis	Open

## Data Availability

Data are available upon reasoning request.

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
