# Peer review of "Averting an Unnecessary Revision of a Roux-en-Y Hepaticojejunostomy by Surgically Creating an Access Point for the Endoscopic Assessment of the Anastomosis: A Report of a Case"

_medicines, 2023, doi:10.3390/medicines10050031_

Round 1

Reviewer 1 Report

1) "relatively high, associated failure": No need for a comma.

2) "While things for malignant obstructions are relatively straightforward in regards to the indications for resection": The use of "things" is very colloquial. Please revise this sentence.

3) "If not properly treated and in a timely fashion": Should be "If not properly treated in a timely fashion"

4) "an unreachable though endoscopy RYHJ": Please correct to "through"

5) Such an approach has already been reported before: "https://pubmed.ncbi.nlm.nih.gov/8553210/" "https://www.ncbi.nlm.nih.gov/pmc/articles/PMC6723649/" and I am sure there are more. Your case report would be much improved if you were to pitch it as a case report and systematic review of the literature, and summarize the key points from all previous published case reports.

6) Also focus on what makes your case different to those that have been previously reported, whether in technical aspects or in patient-related aspects.

Author Response

  1. As per your suggestion, the “comma” has been removed from the phrase “relatively high associated failure” in the abstract section of the revised manuscript.
  2. As per your suggestion, the sentence “While things for malignant obstructions are relatively straightforward in regards to the indication for resection” has been appropriately revised in the introduction section of the revised manuscript.
  3. As per your suggestion, the phrase “if not properly treated and in a timely fashion” has been rephrased to the following: “if not properly treated in a timely fashion” in the introduction section of the revised manuscript.
  4. As per your suggestion, the word “though” has been corrected to the word “through” in the phrase “an unreachable through endoscopy RYHJ” in the introduction section of the revised manuscript.
  5. As per your suggestion, a literature review on the subject has been conducted and has been incorporated in the discussion section of the revised manuscript. A table (Table 1) containing the available reports, including those suggested by you, has been also added in the tables section of revised manuscript. In addition, the key points of the available reports are now discussed in the discussion section of the revised manuscript.
  6. As per your suggestion, comments in regards to what makes our case different to the previously published reported have been added in the discussion section of the revised manuscript

Reviewer 2 Report

The case is interesting. However, I have doubts about the indication of anastomosis revision. A MRCP should have been able to show the lack of biliary anastomosis stenosis. I think that it would be appreciated if you published images of the MRCP. Also, it makes noise to me that the revision was via a laparotomy (I would have chosen laparoscopy as the initial surgical approach, given the idea of performing a procedure the less invasively possible). Finally, in the pictures can be easily appreciated that the surgery was performed via a midline  laparotomy and not a left subcostal, as reported (on the other hand, it'd been a rare incision to choose in order to adequately expose a previous RYHJ).

Author Response

  1. We totally agree with the reviewer that publishing the MRCP images could objectively document and justify our comment in the manuscript that these images were inconclusive in regards to the status of the anastomosis. We tried to retrieve the study through the setting’s intranet persistently. Unfortunately, we did not manage to gain access and these images are no longer available due to system failure affecting the retrieval of imaging data during the patient’s hospitalization period.
  2. We totally agree with the reviewer that a laparoscopic or even a laparoscopic assisted approach would be the ideal approach. However, the preoperative workup determined the hepaticojejunostomy revision as the actual indication for surgery i.e. a challenging procedure performed almost always via laparotomy. We had the false sense, as proven, that a failing anastomosis was the cause of patient’s symptoms. The idea of performing an intraoperative enteroscopy emerged intraoperative, only after the macroscopic assessment and palpation of the anastomosis. As per your suggestion, appropriate comments regarding the decision for an open approach instead of a laparoscopic; have been incorporated in the discussion section of the revised manuscript.
  3. Indeed, the operation was undertaken via a midline incision. By mistake in the original manuscript it was mentioned that a laparotomy via a right subcostal incision was made. In our department, we prefer the subcostal incision for the vast majority of HPB procedures. However, in this certain patient we chose to perform the laparotomy on his previous midline incision due to the presence of a postoperative abdominal wall hernia at the site of the previous incision.

Reviewer 3 Report

congratulations to the authors for their effort. it is indeed an interesting read. 

I would suggest that some English language editing is in order. 

I'm not sure how original this approach is, as I have seen it done before. also, I'm not sure I agree with your assessment that if HIDA scan cannot be performed in your center, surgery should be attempted. why not transfer the patient to a center where the scan can be performed and maybe not submit him to an unnecessary surgery. 

Author Response

  1. As per your suggestion, a native English speaker has performed English language editing of the whole manuscript.
  2. We agree with the reviewer that similar attempts have been made and published in the literature. In this direction, we have incorporated a literature review with all the relevant studies in the field in the discussion section of the revised manuscript. A table containing these studies has been also included in the tables section of the revised manuscript.
  3. We totally agree with the reviewer that the results of a HIDA scan on this patient could, at least theoretically, alter the treatment strategy. However, this certain service is not available in our department. Certainly, transferring the patient to a setting that can perform hepatobiliary scintigraphy studies was an option that was discussed in the department’s MDT meeting. However, the patient’s healthcare insurance logistics precluded this option. Appropriate comments have been incorporated in the discussion section of the revised manuscript.

Round 2

Reviewer 2 Report

I personally would not have revisited the anastomosis, although this is debatable. In the case I’d proceeded with surgery, I would have chosen a laparoscopic approach in order to minimize the patient invasiveness. Otherwise, I concur in that the endoscopic approach to revise a Roux en Y HYA can, like in this presented case, prevent a much more complex surgery such as a roux en Y hepaticoyeyunal re-anastomosis.

Reviewer 3 Report

Thank you for addressing my previous comments.